# External Illumination De-Interfering for Remote Photoplethysmography

## Abstract

Remote photoplethysmography (rPPG), a non-contact technology for extracting physiological signals from facial videos, has drawn increasing interest in the AI community. However, most existing approaches are tailored for idealized studio lighting situations and struggle to generalize to complex real-world scenes. While some studies attempt to mitigate illumination interference by referencing subject-background features, heterogeneous lighting on the face often violates their underlying assumptions, thus limiting further performance gains. To address these challenges, we propose a novel rPPG framework to counteract the adverse effects of complex external illumination on biosignal perception. Considering the unknown and dynamic nature of lighting distributions and their influence on facial imaging variations, we introduce a relative total variation to disentangle global illumination components and preserve high-frequency biosignal transients, while compressing subtle temporal cues within video sequences. This operation enables a contrastive strategy to model facial illumination representations. The captured illumination distribution is then self-supervisedly separated from the original input to yield purified rPPG features. Furthermore, we incorporate a frequency-aware feedforward Transformer to exploit the quasi-periodic nature of pulse waveforms for vital sign estimation. Extensive experiments on multiple public datasets under diverse lighting and motion conditions show that our model achieves competitive performance. The codes are available at: `https://github.com/sachiel0916/dippg/`.

## 1 Introduction

Heart rate (HR) estimation is critical for applications such as face anti-spoofing, disease prevention, identity authentication, and fatigue warning (Choi et al. (2024); Wu et al. (2024)). Traditional electrocardiography and photoplethysmography (PPG) methods require contact sensors, limiting practicality due to complex setup procedures and deployment constraints. Alternatively, vision-based, contactless algorithms (Yang et al. (2023); Wang et al. (2025)) provide scalable and real-time sensing without

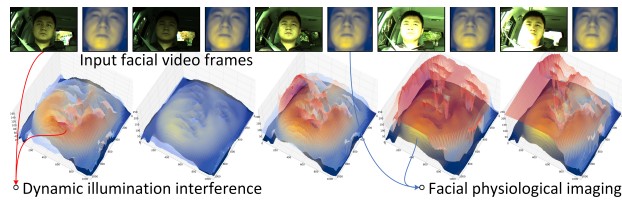
Input facial video frames
Dynamic illumination interference
Facial physiological imaging

Figure 1: Illustration of how intrinsic physiological activity and extrinsic illumination affect facial appearance. Reducing external interference and enhancing biosignal representation are essential for advancing rPPG accuracy.

physical touch, ideal for large-scale scanning in crowded public environments like airports and train stations. Beyond individual-level assessments, these methods can also enable rapid responses during public health emergencies, highlighting their potential in AI-powered physiological sensing.

Facial skin color varies with the blood volume pulsation (BVP), producing periodic patterns linked to HR. This principle underlies remote PPG (rPPG) for HR estimation using computer vision (Haan & Jeanne (2013); Wang et al. (2017)). Recent advances in deep learning have enhanced rPPG detection algorithms (Chen & McDuff (2018); Yue et al. (2025)). For instance, convolutional networks (Li et al. (2023)) identify regions of interest (ROI) through strong biosignal imaging, while Transformer-based models (Shao et al. (2024)) capture long-term dependencies across facial time series to recover physiological signs, thereby continuously improving the accuracy of non-contact HR estimation.

However, pulse-induced facial imaging signals are inherently subtle and readily disturbed by sources of interference. While subject-related disturbances (motion and expressions) have been partially addressed (Shao et al. (2026)), interference caused by external illumination remains underexplored due to its potentially large amplitude variations or stochastic flickering. As the result, most existing rPPG methods perform well indoors but struggle in complex outdoor environments (Wang et al. (2024); Zou et al. (2025a)). As shown in Fig. 1, facial imaging comprises 2 major components: physiological information and illumination variations. Ignoring illumination effects hampers accurate biosignal reconstruction. Effectively separating and suppressing the latter while preserving the former remains a key challenge for practical and robust rPPG applications (Anil et al. (2025); Chen et al. (2025)).

As shown in Fig. 2 (a), rPPG capturing based solely on facial skin area is often affected by external interference, particularly illumination variations, which can significantly contaminate biosignal features. Fig. 2 (b) illustrates that most existing methods address this challenge by leveraging background prior knowledge to model the interference, assuming that the face and background share similar illumination distributions, and apply adversarial (Liu & Yuen (2024)) or contrastive learning (Huang et al. (2025); Shao et al. (2025)) to separate them. However, this mechanism limits the interpretability of the learned representation due to the absence of explicit illumination artifact modeling, thereby restricting the controlled guidance for performance improvement. Moreover, this assumption is not

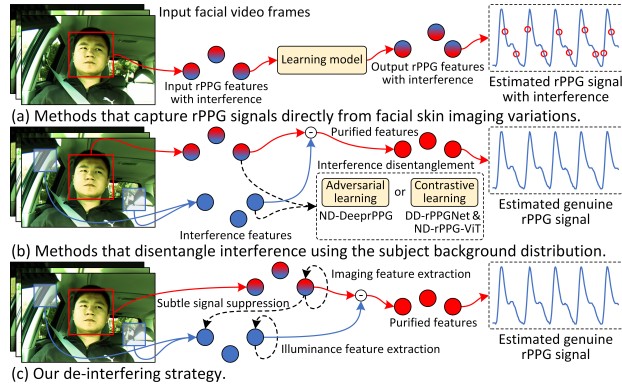

Figure 2: Conventional rPPG pipelines and comparison of de-interfering methods: ours vs. existing approaches.

always valid. In some cases, light may come from varying directions, causing flickering or shadows on the facial surface. Additionally, the face and background also have distinct materials and reflective properties, which can result in uneven lighting, making the background illumination distribution a comparatively less reliable reference and possibly introducing superfluous environmental noise.

To address the challenges above, we propose a novel rPPG framework that explicitly disentangles external interference to improve both robustness and interpretability. Unlike existing methods that suppress interference implicitly in high-dimensional feature spaces, we adopt a "disentangle-then-estimate" strategy (see Fig. 2 (c)). While background distribution is considered, our framework is not entirely reliant on it for interference removal. Instead, we focus on the amplitude and frequency relationships among various imaging components involved in skin color changes, which allows for the extraction of implicit vital signs with quasi-periodic characteristics. Specifically, given the unknown and complex nature of illumination, as well as its potential to interfere with rPPG modeling, we observe that biosignals typically exhibit stable patterns in terms of frequency and amplitude, reflecting a high degree of structural regularity. In contrast, interference introduced by external factors tends to be irregular and lacks such consistency, resembling unstructured signals. Motivated by this distinction between structured and unstructured components, we introduce the relative total variation operation to achieve a global decomposition of the facial imaging. This processing preserves the significant biosignal spikes while effectively suppressing subtle temporal clues. Afterwards, we employ self-supervised learning to represent facial illumination based on the differences in amplitude and frequency between lighting interference and physiological imaging. Finally, the estimated illumination is removed from the raw input video, resulting in purified rPPG features, which are then utilized to inspire a frequency-domain-aware Transformer for HR estimation. We evaluate our method on 3 public datasets: COHFACE (Heusch et al. (2017)), BUAA-MIHR (Xi et al. (2020)), and MR-NIRP (Nowara et al. (2022)), under various lighting situations, motions, and real-world scenarios. Numerous experiments demonstrate that our approach outperforms existing state-of-the-art rPPG methods. The main contributions of this paper are as follows:

- We address the critical challenges that current rPPG methods encounter when dealing with external interference in non-ideal outdoor environments, whereas they are typically limited to deployment in controlled studio settings and static scenarios.

- We introduce an interference decoupling model based on relative total variation of imaging, which reduces reliance on the illumination priors, effectively preventing misinterpretations caused by inconsistencies between background and facial lighting or material properties.

- We propose a hierarchical architecture that integrates interference feature disentanglement with physiological imaging mining, utilizing appearance-structured self-supervised learning to interpret illumination distributions and enhance performance.

- Our network outperforms existing rPPG methods across multiple public datasets and complex scenario verifications, demonstrating its superior robustness and effectiveness.

## 2 RELATED WORKS

*Vision-Based rPPG.* rPPG has attracted growing attention in computer vision for contactless vital signal monitoring. Early approaches, based on biosignal blind source separation and facial imaging chromaticity analysis (Nowara et al. (2018)), were highly sensitive to motions and skin tone changes. Recent advances leverage deep learning to improve the rPPG robustness and performance. Liu et al. (2020; 2023) enhanced dynamic temporal features via inter-frame differences, while Yu et al. (2023) introduced the vision Transformer to capture long-term temporal dependencies. However, their performances degrade under varying scenarios. To improve generalization, Du et al. (2023) synthesized domain noise to reduce distribution gaps, and Lu et al. (2023) promoted broader feature coverage to mitigate activation bias. In terms of network design, Qian et al. (2024) proposed a spatiotemporal dual-path module, and Liu et al. (2025) improved a spike-driven framework. However, they remain inefficient for practical deployment. To address this, Yan et al. (2025) and Luo et al. (2025b) adopted the Mamba to enhance efficiency. In addition, Yue et al. (2023) and Sun et al. (2024) utilized the unsupervised framework to address limited training data. Despite these efforts, most methods are still limited in handling real-world challenges such as complex environments and illumination situations.

*Interference Disentanglement in rPPG.* External illumination interference has long been recognized as a major bottleneck in computer vision-driven rPPG. Initial studies, such as Lee et al. (2015), attempted to suppress the radiance-induced facial artifacts utilizing external fill light in the dark room, but relied heavily on prior contrast distributions and had limited applicability in real-world scenarios. While the rise of deep learning, feature decoupling frameworks emerged. Niu et al. (2020b) encoded non-physiological features and used cross-validation for separation. Chung et al. (2022) disentangled features into domain-specific components, and Qian et al. (2025) applied the diffusion model to decouple vital signals at both frequency and amplitude levels. Despite these advances, they perform remains suboptimal under highly dynamic environments. Background reference modeling has revealed improved robustness. Nowara et al. (2021) introduced an inverse masked attention mechanism to suppress distractions, Liu et al. (2024) employed adversarial learning with noise-prior-based assumptions, while Shao et al. (2025) and Huang et al. (2025) enhanced physiological signal separation via foreground-background similarity. However, they remain lack explicit interpretability of external interference, limiting their ability to guide further performance improvements.

## 3 METHODOLOGY

### 3.1 FACIAL POTENTIAL ILLUMINANCE INFORMATION REPRESENTATION

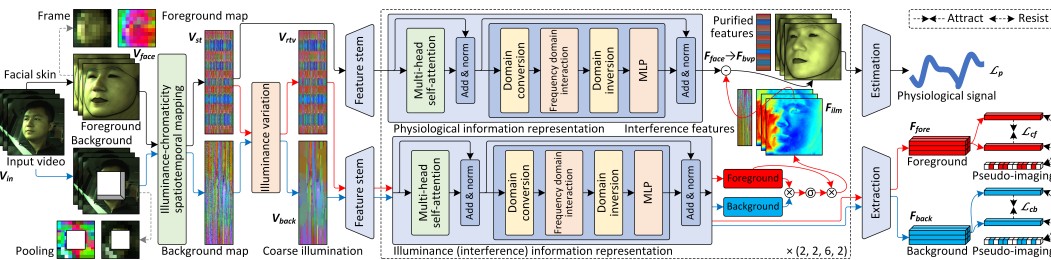

Figure 3: The framework of our de-interfering model employs weight-sharing encoders to extract facial features and capture structural changes in temporal signals. This architecture design effectively decouples external disturbances from physiological cues, thereby enabling more accurate regression.

External illumination interference poses a major conundrum in current rPPG tasks. However, existing methods (Nowara et al. (2021); Liu & Yuen (2024)) predominantly rely merely on background distribution as a reference, which becomes unreliable and susceptible to environmental noise when the face and background exhibit heterogeneous properties. To overcome this issue, we first investigate how to characterize lighting information from unknown distributions without solely depending on the background, then subsequently decouple it from the subtle physiological signal.

To perform this, we first distinguishing the facial skin ROI from the background. For an input video clip: $\mathbf{V}_{\text{in}} \in \mathbb{R}^{3 \times T \times H \times W}$, where $T$ is the number of frames, and $H$ and $W$ are the height and width of each frame, the face is denoted as: $\mathbf{V}_{\text{face}} \in \mathbb{R}^{3 \times T \times H' \times W'}$, where $H'$ and $W'$ are the height and width of identified region. Subsequently, considering the quasi-periodicity of the biosignal, to guide inter-frame skin interactions, we partition $\mathbf{V}_{\text{face}}$ into $S$ non-overlapping rectangle patches per frame, and apply average pooling within each patch. Each frame is then converted into an $L \times 1$ sequence, yielding the spatiotemporal map (STMap) of the facial video, denoted as: $\mathbf{V}_{\text{st}} \in \mathbb{R}^{3 \times L \times T}$. Niu et al. (2020a) have demonstrated that this operation can enhance the temporal dynamic features of weak biosignals, now widely employed in current rPPG methods (Wang et al. (2025); Shao et al. (2026)).

After obtaining the facial STMap $\mathbf{V}_{\text{st}}$, we proceed by extracting its core structured temporal distribution. This step is crucial, as the presence and distribution of external interference during face color changes over time remain unknown. We consider two possible scenarios: 1) external interference affects facial imaging, and 2) there is no severe interference. In the first case, where intense, random, or flickering external interference impacts physiological imaging, our structured temporal extraction strategy can effectively capture illumination changes, isolates the interference, and achieves blind decoupling. In the second case, where the correlated noise is insufficient to obscure the amplitude of biosignal, our strategy enhances the expression of pulse spike information in the waveform, thereby facilitating the quasi-periodic feature learning for rPPG. Afterwards, we apply self-supervised contrastive learning to assess whether the extracted structured time sequence contains interference.

Traditional approaches for extracting structured information from mappings primarily focus on removing fine-grained details, typically through smoothing techniques such as weighted least squares (e.g., linear filtering). Since our strategy involves converting video segments into STMaps, similar operations can be employed within our framework. However, they are inadequate for preserving the spikes of weak physiological signals. In accurate HR measurement, the significance of feature is imbalanced, with spikes at the biosignal's peaks and troughs being particularly significant. Extracting these spike features is critical for mitigating high-frequency noise and improving computational efficiency. Therefore, we can naturally think that depending on the combination of gradient amplitude and total variation, we can preserve temporal spikes as much as possible while denoising. Specifically, the gradient magnitude captures the intensity of local variations within the STMap, which helps identify salient transitions corresponding to spikes. Meanwhile, the total variation factor enforces a global regularization constraint, encouraging smoothness in non-informative regions while maintaining the signal transitions at the peaks.

Among various total variation methods, the TV-$L_2$ model is particularly suitable for extracting structured components from mappings with mixed, unknown, or irregular patterns of structured and unstructured content. It preserves large-scale edges by combining a total variation regularization term with a quadratic penalty to enforce the structural similarity between the input and output. However, it still struggles to differentiate strong structural spikes and local features. To address this limitation, Xu et al. (2012) introduced a relative total variation model. The core insight is that within a local window, domain spikes contribute more to the gradient and share similar directions, whereas features with complex patterns (oscillation) do not exhibit such regularity. This method incorporates a novel regularization term, applies a universal pixel-wise windowed total variation metric, and combines inherent variation within the window to avoid assumptions or manual judgments about local types. Through iterative calculations, it can more effectively distinguish between spikes and stochastic surface details. Taking this benefit, we process $\mathbf{V}_{\text{st}}$, and the resulting $\mathbf{V}_{\text{rtv}}$ can be expressed as:

$$\mathbf{V}_{\text{rtv}}^k = (\mathbf{I} + \lambda \, \mathbf{L}^{k-1})^{-1} \cdot \mathbf{V}_{\text{st}}, \tag{1}$$

where $k$ denotes the iteration step in the variational model, with the regularization term dynamically adjusted based on the mapping content (in this study, $k$ is set to 2), $\lambda$ is a parameter that controls the influence of the regularization term (set to 0.01), $\mathbf{I}$ represents the identity matrix, and $(\mathbf{I} + \lambda \, \mathbf{L}^{k-1})$ is a symmetric positive-definite Laplacian matrix. The discrete gradient is approximated using the forward difference approach, resulting in a sparse five-point Laplacian matrix. Additionally, $\mathbf{L}$ is the

weight matrix computed based on the temporal structure vector $\mathbf{V}'_{\text{rtv}}$ generated from the previous iteration (initialized as $\mathbf{V}_{\text{st}}$). Specifically, $\mathbf{L}$ is given by:

$$\mathbf{L} = \mathbf{C}_s^\top \mathbf{U}_s \mathbf{W}_s \mathbf{C}_s + \mathbf{C}_t^\top \mathbf{U}_t \mathbf{W}_t \mathbf{C}_t, \tag{2}$$

where $\mathbf{C}_s$ and $\mathbf{C}_t$ are the Toeplitz matrices obtained by applying the forward difference approach to the time dimension $t$ and the spatial dimension $s$, respectively, using the discrete gradient operator. The matrices $\mathbf{U}_s$, $\mathbf{U}_t$, $\mathbf{W}_s$, and $\mathbf{W}_t$ are diagonal, with their diagonal elements defined as follows: $\mathbf{U}_s[i,i] = u_{s\,i}$, $\mathbf{U}_t[i,i] = u_{t\,i}$, $\mathbf{W}_s[i,i] = w_{s\,i}$, and $\mathbf{W}_t[i,i] = w_{t\,i}$. To illustrate this process, consider the temporal series within the rectangular region centered on pixel $q$:

$$u_{t\,q} = \left(\mathbf{G}_\sigma * \frac{1}{|\mathbf{G}_\sigma * \partial_t \mathbf{V}'_{\text{rtv}}| + \varepsilon_1}\right)_q, \; w_{t\,q} = \frac{1}{|(\partial_t \mathbf{V}'_{\text{rtv}})_q| + \varepsilon_2}, \tag{3}$$

where $\mathbf{G}_\sigma$ represents the Gaussian filter with standard deviation $\sigma$ set to 5, and $\varepsilon_1$ and $\varepsilon_2$ are small positive constants introduced to avoid division by zero (set to $1\times10^{-3}$ and $2\times10^{-2}$, respectively). In the presence of external interference, the STMap by the relative total variation can act as a reference for lighting changes, mitigating inconsistencies between simulated illumination and background distribution. When external interference is minimal, it can still capture biosignal spikes. The key to detecting interference lies in self-supervised learning based on the specific period and amplitude of physiological signal, which aids in signal decoupling. This process will be discussed in Sec. 4.4.

## 3.2 EXTERNAL DYNAMIC ILLUMINATION DE-INTERFERING

At this stage, we introduce a spatiotemporal feature extraction module designed to encode $\mathbf{V}_{\text{st}}$ and $\mathbf{V}_{\text{rtv}}$, respectively. While $\mathbf{V}_{\text{st}}$ contains all the details of the facial skin imaging, and $\mathbf{V}_{\text{rtv}}$ represents the temporal structure after removing fine-grained information. The resulting feature representations are: $\mathbf{F}_{\text{face}}$ and $\mathbf{F}_{\text{fore}} \in \mathbb{R}^{C \times S \times T}$, where $C$ is the channel dimension and $S$ denotes the compressed tensor space dimension after encoding. These features correspond to the facial characteristics and the foreground features, respectively, which either have an unknown or non-lighting distribution.

At this point, to guide the model in learning temporal sequence features, we design a self-supervised learning mechanism to train the encoder, enabling the network to quickly capture dynamic rhythm characteristics during this phase. To address the issue of inconsistencies reflected light and material differences between the face and background when the background is used as a reference, we perform contrastive learning independently on the face, rather than constructing positive and negative sample pairs in the foreground-background as done in previous rPPG architectures (such as those in Fig. 2 (b)). In the training process, positive samples are $\mathbf{F}_{\text{fore}}$ and its randomly shuffled tensor $\mathbf{F}'_{\text{fore}}$ along the spatial dimension. Negative samples are derived from the $m$ resampled time series tensor $\mathbf{B}_{\text{fore}}$. According to existing research (Jeanningros et al. (2024)), normal HR ranges from 40 to 240 heartbeats per minute (bpm), while video sampling frequencies are typically from 20 to 30 frames per second (fps). To ensure that positive samples correspond to genuine pulse waveforms fall outside the HR confidence interval, we randomly expand or downsample the time series by a factor of $8\times$ to $12\times$. Based on this, we optimize the model using the following loss function:

$$\mathcal{L}_{\text{cf}} = \log\left(\frac{\exp\left(\frac{D(\mathbf{F}_{\text{fore}}, \mathbf{F}'_{\text{fore}})}{\tau}\right)}{\sum_{i=1}^m \left(\exp\left(\frac{D(\mathbf{F}_{\text{fore}}, \mathbf{B}_{\text{fore}_i})}{\tau}\right) + \exp\left(\frac{D(\mathbf{F}'_{\text{fore}}, \mathbf{B}_{\text{fore}_i})}{\tau}\right)\right)} + 1\right), \tag{4}$$

where $D$ represents the mean square error (MSE) between the two tensors with respect to time, and $\tau$ is the temperature hyperparameter which is set to 0.08 as per (Yue et al. (2023)).

Next, we use global illumination intensity for learning. This metric is not entirely based on background distribution, but rather on the trend of its amplitude variation. Therefore, we extract its core structured temporal information while discarding all fine details to avoid introducing additional ambient noise, periodic flickering, and unnecessary oscillations, while still capturing the cross-frame illumination pattern. For the input background, we process it frame by frame, resize it to $H' \times W'$, then apply the STMap transformation and relative total variation to obtain the background STMap: $\mathbf{V}_{\text{back}} \in \mathbb{R}^{3 \times L \times T}$. Subsequently, based on Equ. 5, we employ the encoder to extract background illumination features and handle image variations with random perturbations, and guiding the model for self-supervised temporal distribution mining:

$$\mathcal{L}_{\text{cb}} = \log\left(\frac{\exp\left(\frac{D(\mathbf{F}_{\text{back}}, \mathbf{F}'_{\text{back}})}{\tau}\right)}{\sum_{i=1}^m \left(\exp\left(\frac{D(\mathbf{F}_{\text{back}}, \mathbf{B}_{\text{back}_i})}{\tau}\right) + \exp\left(\frac{D(\mathbf{F}'_{\text{back}}, \mathbf{B}_{\text{back}_i})}{\tau}\right)\right)} + 1\right), \tag{5}$$

where $\{\mathbf{F}_{\text{back}}, \mathbf{F}'_{\text{back}}\}$ and $\{\mathbf{B}_{\text{back}_1}, ..., \mathbf{B}_{\text{back}_i}\}$ are the augmented positive samples and the negative samples, respectively. The acquisition manner follows the same procedure as described in Equ. 4.

Subsequently, we extract the similar distributions from both the facial and background core temporal structure tensors ($\mathbf{F}_{\text{fore}}$ and $\mathbf{F}_{\text{back}}$), and compute their respective similarity matrices, which are then regularized to emphasize the most relevant temporal relationships. The regularized similarity matrices are subsequently multiplied by the facial temporal pattern feature matrix, yielding the illumination distribution features $\mathbf{F}_{\text{ilm}}$:

$$\mathbf{F}_{\text{ilm}} = \text{Softmax}\Big(\text{PSD}(\mathbf{F}_{\text{back}}) \cdot \big(\text{PSD}(\mathbf{F}_{\text{fore}})\big)^{\mathsf{T}}\Big) \cdot \mathbf{F}_{\text{fore}}, \tag{6}$$

where PSD represents the power spectral density processing applied to the features. Ultimately, the purified rPPG features used for regressing physiological signals are expressed as: $\mathbf{F}_{\text{bvp}} = \mathbf{F}_{\text{face}} - \mathbf{F}_{\text{ilm}}$. In comparison with existing decoupling methods, the proposed approach allows for contrastive analysis, thereby enabling more informed visual inspection, guiding targeted model improvements, and contributing to a more robust evaluation of the rPPG detection results.

### 3.3 Physiological Signal Regression and Model Optimization

Once the feature $\mathbf{F}_{\text{bvp}}$ is calculated, physiological signal regression is performed using the estimator. This step enables the mapping of the extracted features to specific waveforms, facilitating accurate HR prediction. Therefore, we impose a constraint based on the negative Pearson's correlation:

$$\mathcal{L}_{\text{p}} = 1 - \frac{T \sum_{i=1}^{T} P_{\text{s}_i} P_{\text{g}_i} - \sum_{i=1}^{T} P_{\text{s}_i} \sum_{i=1}^{T} P_{\text{g}_i}}{\sqrt{\Big(T \sum_{i=1}^{T} P_{\text{s}_i}^2 - (\sum_{i=1}^{T} P_{\text{s}_i})^2\Big)\Big(T \sum_{i=1}^{T} P_{\text{g}_i}^2 - (\sum_{i=1}^{T} P_{\text{g}_i})^2\Big)}}, \tag{7}$$

where $P_{\text{s}}$ denotes the model-predicted biosignal, while $P_{\text{g}}$ is the corresponding ground truth (GT).

Regarding the overall network framework, as illustrated in Fig. 3, we design our architecture with the Swin Transformer (Liu et al. (2022)) as the encoder backbone. The video segment input initially passes through a feature stem for preliminary spatiotemporal integration, thereby enhancing low-level representations. The encoder consists of twelve Swin Transformer modules, organized into four hierarchical Swin stages with

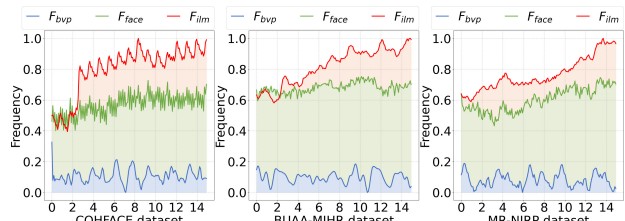

Figure 4: Distribution of imaging feature frequencies.

2, 2, 6, and 2 modules per stage. Spatial downsampling (by a factor of $2\times$) is executed in each stage while maintaining the temporal dimension. Each stage alternately stacks window-based multi-head self-attention and shifted-window multi-head self-attention, enabling hierarchical feature extraction while balancing local pulse spike modeling and global context integration. The attention head counts for these stages are 3, 6, 12, and 24, respectively, with a fixed window size of 8. Compared to convolutional architectures, our STMap-oriented model more effectively captures long-range dependencies and cross-frame contextual information in time series. The shifted window mechanism further facilitates interaction between local and global features, which is key for modeling the dynamic and weak temporal signals associated with rPPG regression tasks.

As shown in Fig. 4, the temporal-frequency characteristics of different feature imaging contents exhibit variability. To more appropriately capture the unique quasi-periodic nature of cardiac pulses and distinguish them from noise, we replace the standard feedforward module in the vision Transformer with a dual-domain modeling module that integrates information from both the temporal and frequency domains. This module consists of three key components: domain conversion, frequency domain interaction, and domain inversion. Specifically, the domain conversion module applies the fast Fourier transform (FFT) to project time-domain features into the frequency domain, thereby explicitly incorporating periodicity and spectral characteristics. Then, the domain interaction module captures and reweights frequency components via a fully connected layer, emphasizing structural relationships among key frequencies. Finally, the domain inversion module applies the inverse FFT to project the enhanced frequency features back into the temporal domain, ensuring their effective

Table 1: Comparative results (in bpm and dB). Here ↓ indicates that lower is better, ↑ is vice versa, * is unsupervised methods, the best result is **bolded**, and the second-best result is underlined.

| rPPG Methods | | COHFACE (Heusch et al. (2017)) | | | | BUAA-MIHR (Xi et al. (2020)) | | | | MR-NIRP (Nowara et al. (2022)) | | | |
|---|---|---|---|---|---|---|---|---|---|---|---|---|---|
| | | MAE↓ | RMSE↓ | $\rho$↑ | SNR↑ | MAE↓ | RMSE↓ | $\rho$↑ | SNR↑ | MAE↓ | RMSE↓ | $\rho$↑ | SNR↑ |
| CHROM(Haan & Jeanne (2013)) | | 11.22 | 15.83 | 0.31 | –5.91 | 6.09 | 8.29 | 0.51 | –4.21 | 14.52 | 17.41 | 0.18 | –3.60 |
| POS(Wang et al. (2017)) | | 11.65 | 15.14 | 0.35 | –3.21 | 5.04 | 7.12 | 0.63 | 0.81 | 12.75 | 15.36 | 0.34 | –0.31 |
| AutoSparsePPG(Nowara et al. (2022)) | | 9.54 | 13.82 | 0.33 | –2.93 | 5.36 | 7.46 | 0.63 | –2.90 | 5.67 | 10.62 | 0.68 | –3.56 |
| CNN-Based | DeepPhys(Chen & McDuff (2018)) | 6.79 | 12.34 | 0.31 | 1.48 | 4.78 | 6.74 | 0.69 | 1.58 | 13.22 | 18.39 | 0.43 | –2.39 |
| | TS-CAN(Liu et al. (2020)) | 7.65 | 10.90 | 0.40 | 2.14 | 4.84 | 6.89 | 0.68 | –0.16 | 12.70 | 18.03 | 0.47 | 2.95 |
| | DualGAN(Lu et al. (2021)) | 6.79 | 8.56 | 0.68 | 2.22 | 3.41 | 5.23 | 0.84 | 3.06 | 8.00 | 12.18 | 0.71 | 4.31 |
| | PFE-TFA(Li et al. (2023)) | 6.68 | 9.38 | 0.66 | 1.87 | 1.29 | 2.65 | 0.91 | 3.94 | 5.34 | 8.92 | 0.73 | 3.95 |
| | NEST(Lu et al. (2023)) | 7.01 | 11.41 | 0.64 | 2.15 | 2.88 | 4.69 | 0.89 | 4.36 | 3.61 | 7.32 | 0.82 | 4.12 |
| | Contrast-Phys+(Sun & Li (2024))* | 7.52 | 15.23 | 0.62 | 2.03 | 4.64 | 6.51 | 0.73 | 3.11 | 6.70 | 11.21 | 0.63 | 2.70 |
| | rPPG-HiBa(Wang et al. (2024)) | - | - | - | - | 2.45 | 3.28 | **0.98** | - | - | - | - | - |
| | ND-DeeprPPG(Liu & Yuen (2024)) | 5.27 | **6.91** | 0.77 | 3.24 | 0.58 | 1.81 | 0.95 | 7.28 | 3.47 | 6.54 | 0.85 | 4.73 |
| | DD-rPPGNet(Huang et al. (2025))* | 8.54 | 8.86 | 0.46 | - | - | - | - | - | 13.93 | 15.14 | 0.18 | - |
| Transformer | EfficientPhys(Liu et al. (2023)) | 5.70 | 8.13 | 0.74 | 2.59 | 1.43 | 4.98 | 0.93 | 4.95 | 3.67 | 12.28 | 0.81 | 4.07 |
| | PhysFormer++(Yu et al. (2023)) | 5.35 | 7.72 | 0.76 | 3.88 | 0.93 | 1.66 | 0.94 | 5.33 | 3.56 | 7.59 | 0.83 | 4.15 |
| | Spiking-PhysF.(Liu et al. (2025)) | 5.01 | 7.99 | 0.80 | 2.83 | 2.12 | 5.08 | 0.90 | 6.76 | 3.62 | 7.39 | 0.85 | 4.88 |
| | RhythmFormer(Zou et al. (2025a)) | 5.42 | 8.26 | 0.73 | 4.09 | 0.67 | 1.57 | 0.94 | 7.40 | 3.44 | 6.72 | 0.82 | 5.56 |
| | ND-rPPG-ViT(Shao et al. (2025)) | 5.03 | 8.25 | **0.82** | 3.87 | 0.53 | 1.20 | 0.95 | 8.58 | 2.69 | 6.03 | 0.87 | 5.63 |
| Mamba | PhysMamba(TD)(Luo et al. (2024)) | 6.04 | 8.30 | 0.71 | 2.96 | 1.09 | 1.94 | 0.86 | 4.83 | 3.89 | 7.92 | 0.79 | 2.80 |
| | RhythmMamba(Zou et al. (2025b)) | 5.48 | 8.03 | 0.73 | 3.74 | 0.96 | 1.82 | 0.90 | 7.10 | 3.31 | 6.36 | 0.84 | 5.53 |
| | PhysMamba(SSD)(Yan et al. (2025)) | 4.97 | 8.22 | 0.72 | 3.23 | 0.89 | 1.89 | 0.88 | 7.09 | 3.60 | 6.45 | 0.82 | 3.59 |
| Ours | | **4.89** | 7.94 | **0.82** | **4.85** | **0.50** | **1.16** | **0.98** | **9.37** | **2.58** | **5.90** | **0.88** | **5.75** |

fusion with the original rPPG information. This feedforward strategy preserves the nonlinear transformation capabilities of the vanilla architecture while significantly enhancing expressiveness and predictive performance for periodic time series data, especially physiological signals.

Additionally, the feature regression module adopts a convolutional architecture consisting of two groups of submodules, each containing two 3×3 convolutional layers, followed by the ReLU activation and batch normalization. A linear projection layer then reduces spatial dimensions and converts high-dimensional features into time-series form for regression of the target physiological signal (output $P_s$). The overall loss function $\mathcal{L}_{\text{total}}$ is defined as follows: $\mathcal{L}_{\text{total}} = \alpha\,\mathcal{L}_p + \beta\,(\mathcal{L}_{cf} + \mathcal{L}_{cb})$, where $\alpha$=0.5 and $\beta$=1 in our setting following the ablation studies (Sec. 4.4).

# 4 EXPERIMENTS

## 4.1 IMPLEMENTATION DETAILS

We train our network and conduct extensive experiments on the publicly recognized and accessible rPPG datasets: COHFACE, BUAA-MIHR, and MR-NIRP, which serve as important benchmarks for evaluating remote vital sensing performance in complex settings. Specifically, the COHFACE dataset in-

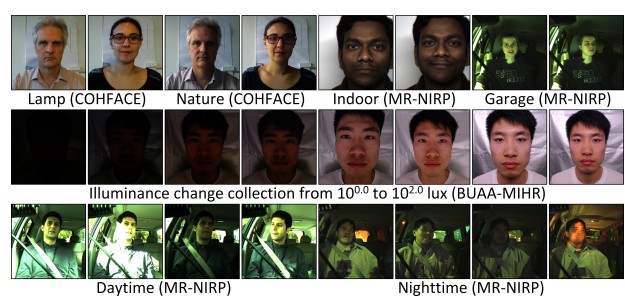

Figure 5: Illustration of sample changes in rPPG datasets.

cludes recordings captured in indoor environments characterized by insufficient natural light, uneven illumination conditions, and supplemental artificial lighting from ceiling lamps. The BUAA-MIHR dataset provides eleven controlled lighting scenarios, with illumination intensities ranging from level $10^{0.0}$ to $10^{2.0}$ lux. The MR-NIRP dataset is a large-scale collection from varied real-world outdoor scenarios, such as actual driving, encompassing significant lighting variations, and day-night transitions. Their representative participants and corresponding changes are illustrated in Fig. 5.

Given the differences in video frame rates and biosignal label frequencies across datasets, we uniformly interpolate them to 25 fps. The open-source face_recognition module is used to segment the facial ROI box and background area. Based on this, the temporal dimension ($T$) of each STMap is set to 320, and the spatial scale ($L$) is set to 64. Following (Yue et al. (2023)), the number of negative samples ($m$) in our illumination self-supervised representation constraint is define to 4. Our model is implemented using the PyTorch framework and runs on a system equipped with four NVIDIA RTX 4090 GPUs. We employ the AdamW optimizer and conduct 100 training epochs, starting with an initial learning rate of $1\times10^{-5}$, which is adjusted to $0.5\times10^{-5}$ after the 50th epoch. Notably, to prevent cross-contamination during training and testing, we divide the samples in each dataset according to scenario, subject, and illumination, randomly selecting 3/4 of the samples for training and the remaining 1/4 for testing, rather than partitioning them by random STMap-level.

## 4.2 COMPARISON WITH STATE-OF-THE-ART rPPG APPROACHES

We train and test each dataset independently, and combining MR-NIRP with its indoor supplementary set (Nowara et al. (2018)) to enhance sensing robustness. Since the network outputs temporal waveform sequences, we utilize the open-source rPPG toolbox[1] to calculate and statistically analyze HR values. For evaluation, we employ the mean absolute error (MAE), root MSE (RMSE),

Table 2: RMSE across different scenes and motion states.

| rPPG Methods | Scenarios | | | | | Movements | |
| | COHFACE | | MR-NIRP | | | | |
| | Lamp | Nature | Day | Night | Garage | Motion | Still |
|---|---|---|---|---|---|---|---|
| AutoSparsePPG | 11.58 | 17.03 | 15.92 | 16.23 | 2.90 | 17.69 | 16.10 |
| PhysFormer++ | 6.63 | 11.93 | 9.19 | 8.75 | 3.99 | 10.35 | 7.05 |
| ND-DeeprPPG | **5.67** | 11.59 | 7.69 | 8.02 | 3.50 | 7.46 | 6.28 |
| RhythmFormer | 6.49 | 10.30 | 8.18 | 9.23 | 3.82 | 7.76 | 6.34 |
| ND-rPPG-ViT | 7.35 | 9.66 | 5.45 | 7.51 | 2.15 | 7.30 | 5.74 |
| Ours | 5.93 | **9.40** | **5.34** | **7.48** | **2.05** | **6.95** | **5.58** |

and Pearson's correlation coefficient ($\rho$) of the HR values, along with the signal-to-noise ratio (SNR) of the pulse waveforms. Based on these metrics, we compare our results with the most representative and current state-of-the-art rPPG approaches, as summarized in Tab. 1.

The experimental results on the COH-FACE, BUAA-MIHR, and MR-NIRP datasets demonstrate that the proposed approach achieves competitive performance across diverse situations. Crucially, on BUAA-MIHR and MR-NIRP datasets, which are characterized by the complex, varying, and extreme lighting conditions, our targeted external illumination de-interfering solution significantly outperforms existing rPPG algorithms across multiple quantitative evaluation metrics. On the COHFACE dataset, however, our approach exhibits

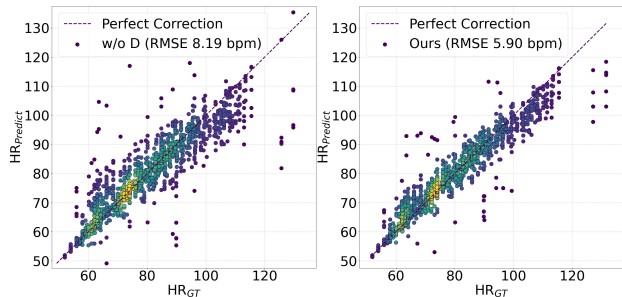

Figure 6: De-interfering performance on MR-NIRP.

the slightly higher RMSE compared to ND-DeeprPPG (Liu & Yuen (2024)). This can be attributed to the relatively controlled and simplistic video conditions of COHFACE, where limited variability and stable lighting reduce the performance disparity among different methods. Nevertheless, further analysis (see Tab. 2) reveals that our method performs notably well under low-illumination scenarios within COHFACE, underscoring its robustness in challenging lighting environments.

To moreover validate our performance across diverse scenes, we conduct additional experiments on COHFACE and MR-NIRP datasets, specifically assessing the impact of varying lighting conditions and subject movements. These evaluations provide a more comprehensive evaluation of the robustness of our algorithm in complex and real-world settings. As presented in Tab. 2, our approach consistently outperforms base-

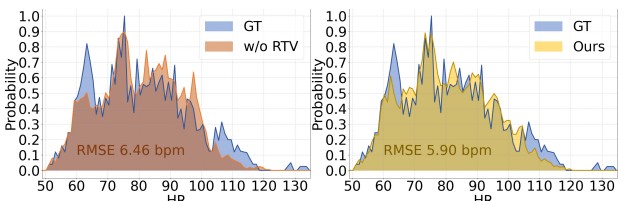

Figure 7: Variational approach results on MR-NIRP.

line frameworks, particularly under low-light and high-motion conditions, demonstrating its strong adaptability and effectiveness. This also explains the slightly suboptimal performance of our method relative to ND-DeeprPPG Liu & Yuen (2024) in the RMSE metric on the COHFACE dataset.

## 4.3 PERFORMANCE ANALYSIS AND DE-INTERFERING VISUALIZATION

We discuss the architectural design in stages. First, we evaluate the effectiveness of the proposed de-interfering strategy. To this end, we design two rPPG networks: one incorporating the disentanglement framework and one without it. Both models are trained and tested on the MR-NIRP dataset, and their HR estimation results are visualized using scatter plots, as shown in Fig. 6. We also report the corresponding RMSE values to provide a quantitative comparison. The scatter plot of our full

---

[1]https://github.com/PHUSELab/pyVHR/

model (right) presents a stronger correlation with the GT, and closely follows the identity line. In contrast, the network without the de-interfering framework (left) exhibits larger deviations, particularly in boundary cases (extremely large or small points). This demonstrates that our design not only improves overall estimation accuracy but also enhances robustness under challenging conditions.

Next, we evaluate the effect of incorporating the relative total variation to the STMap. Using the MR-NIRP dataset, we compare algorithms with (right) and without (left) this component. As seen in Fig. 7, we plot the distribution of GT, along with predicted HR distributions from both methods, and report RMSE. It is evident that the variational model produces results closer to the GT, especially under challenging conditions.

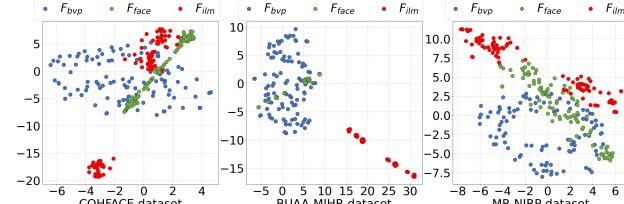

Figure 8: Visualization of frequency domain features.

We extract features from the final layer of the encoder and compute the frequency characteristics of $\mathbf{F}_{\mathrm{bvp}}$, $\mathbf{F}_{\mathrm{face}}$, and $\mathbf{F}_{\mathrm{ilm}}$ separately. These features are then visualized using t-SNE, as shown in Fig. 8. Allowing us to assess their distributions in a low-dimensional space and evaluate the effectiveness of rPPG feature extraction and external interference disentanglement.

### 4.4 EFFECTIVENESS ANALYSIS AND ABLATION STUDIES

We conduct a series of ablation studies on the COHFACE dataset to evaluate the impact of key algorithmic choices and network parameter settings, using MAE as the evaluation criterion. As illustrated in Fig. 9, the experiments are divided into three parts: 1) the effect of different temporal and spatial scales in the STMap (left), where temporal scales ($T$) of 80, 160, 320, and 480 frames, and spatial sizes ($L$) of 32, 64,

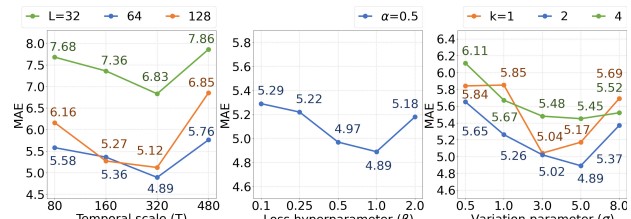

Figure 9: Analysis of our parameter configurations.

and 128 pixels are learned; 2) the impact of varying the loss hyperparameter $\beta$ while keeping $\alpha$ fixed (middle); and 3) the effect of variational model configurations (right), including the number of iterations ($k$), and the core Gaussian parameter ($\sigma$). They validate our model under various settings.

We compare the efficiency of our model against several existing approaches in terms of parameter count, floating point operations per second (FLOPs), and inference time, all benchmarked on an RTX 4090 GPU. Additionally, we report their RMSE indexes on MR-NIRP. As shown in Tab. 3, our method reveals an overall competitive advantage across

Table 3: Computational cost and performance comparison.

| rPPG Methods | Parameters | FLOPs | RTX 4090 GPU | RMSE↓ |
|---|---|---|---|---|
| TS-CAN | 3.91 M | 110.15 G | 5.52 ms | 18.03 bpm |
| PhysFormer++ | 9.79 M | 49.85 G | 217.07 ms | 7.59 bpm |
| ND-DeeprPPG | 6.05 M | 320.08 G | 29.87 ms | 6.54 bpm |
| RhythmFormer | 3.25 M | 38.49 G | 29.49 ms | 6.72 bpm |
| ND-rPPG-ViT | 6.03 M | 55.04 G | 24.64 ms | 6.03 bpm |
| Ours | 5.97 M | 55.20 G | 23.12 ms | 5.90 bpm |

both accuracy and computational efficiency, highlighting its suitability for real-time applications.

## 5 CONCLUSION

Our proposed method introduces a principled disentanglement framework for robust rPPG estimation under complex conditions. By leveraging relative total variation to suppress global illumination while preserving critical subtle physiological cues, our method enables self-supervised learning of facial illumination representations and effective recovery of clean biosignals. Extensive experiments across diverse real-world scenarios and benchmark datasets demonstrate that our method consistently outperforms the existing state-of-the-art rPPG approaches, offering a significant advancement toward practical, unconstrained remote physiological signal and vital sign sensing.

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
