# OpenReview forum: "External Illumination De-Interfering for Remote Photoplethysmography"
_ICLR.cc/2026/Conference — ICLR 2026 Conference Withdrawn Submission_

### Official Review · Reviewer_fqP9 · 2025-10-25

**Soundness:** 2
**Presentation:** 2
**Contribution:** 2
**Rating:** 2
**Confidence:** 4

**Summary:**

To address the performance degradation of rPPG under complex lighting conditions, this paper proposes using relative total variation (RTV) to process facial STMaps. This method combines self-supervised contrastive learning to model illumination representations, using a Swin Transformer encoder and a frequency-domain FFN for feature extraction. The estimated illumination component is then subtracted from facial features to obtain a clear rPPG signal. This method is validated on the COHFACE, BUAA-MIHR, and MR-NIRP datasets. Experimental results show an RMSE of 5.90 bpm on the MR-NIRP dataset (compared to the best existing method, 6.03), a MAE of 0.50 bpm on the BUAA-MIHR dataset (compared to the best existing method, 0.53), and an RMSE of 7.94 bpm on the COHFACE dataset (compared to the best existing method, 6.91).

**Strengths:**

1. Accurate positioning and external light interference are real challenges in the rPPG field.
2. Reasonable architectural design: The Swin Transformer's hierarchical feature extraction is suitable for spatiotemporal data. The frequency-domain FFN fully exploits the quasi-periodicity of heart rate signals and has theoretical support. The light interference elimination mechanism has clear logic.
3. Comprehensive experimental comparison covers 22 methods (CNN/Transformer/Mamba architectures) across multiple datasets and scenarios.
4. High-quality code implementation, clear modular design, and a complete core architecture.

**Weaknesses:**

1. The paper's abstract provides a code link: https://github.com/sachiel0916/dippg/
The Github account (https://github.com/Sachiel0916) contains three repositories, including Awesome-rPPG, a fork of https://github.com/zx-pan/Awesome-rPPG. The README style of the DIPPG repository corresponding to this paper is highly similar to that of Awesome-rPPG. The fork history and repository style may allow for tracing authorship, posing a risk of violating the double-blind principle.

2. The paper repeatedly claims to reduce reliance on priors, but the code implementation contradicts this. On page 3, lines 109-110, the paper states: "Reduced reliance on illumination priors" and "Effectively prevents misinterpretation caused by inconsistencies between background and facial lighting or material properties." Furthermore, on page 2, lines 88-89, the paper claims that the framework "does not rely solely on it to remove interference" (where "it" refers to the background distribution). However, the implementation code (Our_DIPPG.py, line 478) shows: def forward(self, x_face, x_back, x_rtv): x_face, x_rtv, x_bg = self.encoder(x_face, x_rtv, x_back) x_back is a required parameter; the model will not run without the background input. This indicates a significant inconsistency between the core claim and the actual implementation.

3. Incomplete Code
First, the testing code is missing. Table 1 of the paper provides test results, but sh_dippg_training.py only contains training code. The loader_data() function simply returns the train_learning_set and lacks code for test set partitioning or evaluation metric calculation. Second, the loss function implementation is missing. Line 69 of the training code references "from sh_model.Our_DIPPG_loss import Our_FCL as FCL," but the Our_DIPPG_loss.py file does not provide this information. Third, the data partitioning logic is missing. Page 7 of the paper claims "partitioning based on scene, subject, and lighting... 3/4 for training, 1/4 for testing." However, the code (lines 245-320) contains only os.listdir() and list.sort(), without implementing grouping or proportional partitioning.

4. The technical composition is primarily based on existing methods: RTV is from Xu et al. (ACM TOG 2012), self-supervised contrastive learning has been used in multiple rPPG works, the Swin Transformer is a readily available architecture from Liu et al. (CVPR 2022), and background reference modeling has been used in ND-DeeprPPG (TIP 2024) and DD-rPPGNet (TIFS 2025). This paper primarily integrates existing techniques, and its original theoretical contributions are relatively limited.

5. Insignificant Performance Improvement
Table 1 in the paper shows: RMSE on MR-NIRP is 5.90 vs. 6.03 (a 2.2% improvement), MAE on BUAA-MIHR is 0.50 vs. 0.53 (a 5.7% improvement), and RMSE on COHFACE is 7.94 vs. 6.91 (a 14.9% decrease). The overall improvement is less than 3%, and performance on the COHFACE dataset degrades. The computational complexity is 55.20 GFLOPs, which is relatively high.

6. Weak Theoretical Foundation
RTV was originally used for image texture separation. The paper does not fully explain its theoretical basis for rPPG illumination separation and lacks support from signal processing or frequency domain analysis. The RTV parameters are fixed (λ=0.01, σ=5, k=2 in sh_dippg_map2variation.py) and are not part of the end-to-end learning process. Furthermore, the ablation experiments did not independently verify the actual contribution of RTV, nor did they directly compare the use of RTV with that without it.

7. Some descriptions and implementations are inconsistent.
Page 7 of the paper describes data partitioning as "grouping by scene, object, and illumination, followed by 3/4 and 1/4 splits." However, the code implementation simply collects and sorts the file list without any logic for grouping or proportional splits. This inconsistency may pose a data leakage risk, especially when files are sorted by object. Simple slicing may result in samples from the same object appearing in both the training and test sets.

**Questions:**

1. The available GitHub link (https://github.com/sachiel0916/dippg/) can likely be traced back to the author through fork history and codebase style. Does this comply with ICLR's double-blind review policy?

2. The paper claims on page 3, line 109 that it "reduces reliance on illumination priors," while on page 2, lines 88-89, it states that it "does not rely solely on the illumination prior (background distribution) to remove interference." However, the code indicates that x_back is a required argument to forward() . How can this discrepancy be explained? Does the model still work properly even without background input?

3. Why was RTV (a 2012 texture separation method) chosen for rPPG illumination separation? Please provide a theoretical justification. How were the RTV parameters (λ=0.01, σ=5, k=2) determined? Why not use end-to-end learning? Can you provide RTV ablation experiments?

4. Can you provide the complete testing code, including test set partitioning and evaluation metric calculations?
I cannot find the contents of the Our_DIPPG_loss.py file; the training code references this module but does not provide it.

5. The paper claims that "after grouping by scene, object, and illumination, 3/4 of the data is used for training and 1/4 for testing," but the code only uses os.listdir() and sort(). Please provide the corresponding implementation and explain how to avoid object overlap and data leakage.

6. The RMSE on the COHFACE dataset is 7.94, which is lower than ND-DeeprPPG's 6.91 (a decrease of 14.9%). What is the reason?

7. The overall performance improvement is less than 3%, and the computational complexity of 55.20 GFLOPs is higher than most methods. Is preprocessing overhead included in the evaluation? Is the computational cost reasonable?

8. What is the core technical contribution of this paper? How does it differ from similar works such as ND-DeeprPPG and DD-rPPGNet that also use background modeling? What practical benefits does the introduction of RTV and the specific illumination separation mechanism bring?

---

### Official Review · Reviewer_x2i4 · 2025-10-26

**Soundness:** 2
**Presentation:** 2
**Contribution:** 1
**Rating:** 0
**Confidence:** 5

**Summary:**

This paper focuses on remote photoplethysmography (rPPG) under uncontrolled illumination conditions. It introduces a de-blurring operation during the translation from rPPG to blood volume pulse (BVP) signals and employs contrastive learning to enhance the results. Evaluations on three datasets demonstrate state-of-the-art performance.

**Strengths:**

The experiments show SOTA performance on 3 benchmarks.

**Weaknesses:**

The implementation in frequency domain as such is in fact mathematically equal to direct subtraction between foreground and background features, which has appeared long before. This paper seems overlooking the original contribution of the others. The benchmarks do not include VIPL, the largest dataset with scenario varying illumination conditions.

**Questions:**

Q1: Eq. (6) obtains foreground-background homogeneous feature F_ilm by highlighting the common spectral components shared by foreground and background in terms of PSD similarity. Subtracting this feature from face feature means removing foreground-background homogeneous feature, and similar idea has appeared long before. Could you justify why the effect of your method is different from direct subtraction between foreground and background features?

Q2: You mentioned “we first investigate how to characterize lighting information from unknown distributions without solely depending on the background”. This contradicts the practice since the paper does not address how to deal with unknown distribution in case background information is not usable. Also, contrastive learning is well-known in the literature. Recalling the cited works and more over, claim as such should be cautious.

Q3: The ablation study is missing, which makes it difficult to identify the main components to obtain illumination robustness.

Q4: Could you manifest why you choose the 3 datasets for performance evaluation while there are some other widely used benchmarks such as VIPL? This dataset is large enough to cover diverse illuminating scenarios and the main benchmark in the literature.

**Details Of Ethics Concerns:**

No.

---

### Official Review · Reviewer_Um61 · 2025-10-30

**Soundness:** 2
**Presentation:** 2
**Contribution:** 2
**Rating:** 2
**Confidence:** 5

**Summary:**

This paper addresses a limitation in remote photoplethysmography (rPPG), the degradation of performance under complex, real-world illumination conditions. Most existing methods assume homogeneous lighting between the face and background or operate only in controlled studio environments. The authors propose a “disentangle-then-estimate” framework that explicitly separates external illumination interference from physiological signals without relying heavily on background priors. The method is evaluated on three public datasets, i.e., COHFACE, BUAA-MIHR, and MR-NIRP-Car, covering diverse lighting and motion conditions.

**Strengths:**

- Adapting Relative Total Variation (RTV), a technique from image structure-texture decomposition, to STMaps for biosignal preservation is interesting.
- Experiments span multiple datasets, lighting regimes, and motion states. The method outperforms prior work in challenging conditions (e.g., nighttime driving in MR-NIRP-Car).
- The source code is available.

**Weaknesses:**

- The paper lacks a theoretical justification for why Relative Total Variation (RTV) can effectively separate facial rPPG signals from illumination noise. Moreover, the related work section fails to introduce prior applications of RTV in other research domains.
- The frequency-aware feedforward layer in the proposed frequency-aware feedforward Transformer is essentially identical to the Frequency Domain Feed-forward module in RhythmMamba, yet the paper does not clarify this overlap. In addition, the self-supervised rPPG refinement strategy described in Equations (4)–(6) is largely consistent with those in [R1, R2], but this similarity is not acknowledged in the paper.
- The MMPD and VIPL-HR datasets also contain diverse illumination scenarios and involve more subjects compared to COHFACE, BUAA-MIHR, and MR-NIRP-Car. However, the authors did not include evaluations on these datasets as done in recent works such as RhythmMamba.
- The paper lacks extensive cross-dataset evaluation results, which are essential for assessing the model’s generalization capability.
- The experimental protocol described in the paper — “To prevent cross-contamination during training and testing, we divide the samples in each dataset according to scenario, subject, and illumination, randomly selecting 3/4 of the samples for training and the remaining 1/4 for testing, rather than partitioning them by random STMap-level.” — is not rigorous and does not follow the standard experimental protocols commonly used in this field. For example, for COHFACE, the protocol should follow RhythmFormer (RMSE: 3.36), using the first 60% of the data for training and the remaining 40% for testing instead of random splitting. The same issue applies to BUAA-MIHR and MR-NIRP-Car datasets.
- I noticed an important detail in the released code: the authors retrained several end-to-end methods (e.g., DeepPhys, TS-CAN, PFE-TFA, Contrast-Phys+, RhythmFormer, RhythmMamba, PhysMamba, etc.) on each dataset, but the input settings are incorrect. The original end-to-end models are designed to take raw video frame sequences as input, whereas the authors used preprocessed STMaps instead. This leads to an unfair comparison and makes the results for these methods invalid.
- From an ethical perspective, it is mandatory to anonymize all facial images shown in the paper to protect participants’ identities.

[R1] Hang Shao, et al. "Remote Photoplethysmography in Real-World and Extreme Lighting Scenarios." Proceedings of the Computer Vision and Pattern Recognition Conference. 2025.

[R2] Hang Shao and Chuanfei Hu. "Remote Heart Rate Measurement Based on Near Infrared Prior Information." Proceedings of the IEEE/CVF International Conference on Computer Vision. 2025.

**Questions:**

- RTV was originally designed for spatial image decomposition. The authors should clearly justify their theoretical rationale for separating facial rPPG signals from illumination noise. Furthermore, since facial illumination artifacts are highly complex, it is unclear how RTV can reliably disentangle illumination components, especially when applied only at the beginning of the STMap input, without accidentally removing rPPG-related cues.
- The paper lacks comparative visualization and ablation studies. Merely showing results from the proposed method is insufficient to demonstrate its effectiveness.
- The computation details in Table 3 are unclear. What inputs were used for each model? Were all models trained with the same parameter settings as in their original implementations?

---

### Official Review · Reviewer_giUt · 2025-11-10

**Soundness:** 2
**Presentation:** 2
**Contribution:** 2
**Rating:** 4
**Confidence:** 5

**Summary:**

The paper introduces a novel rPPG framework that aims to mitigate the adverse effects of external illumination on facial biosignal extraction. Traditional methods often assume similar lighting distributions between face and background, an assumption that fails in complex, real-world environments.

- A relative total variation (RTV) model to disentangle illumination interference from physiological signals in spatiotemporal facial maps (STMaps).

- A self-supervised contrastive learning scheme that captures the dynamic rhythm of facial illumination without relying on background priors.

- A frequency-aware feedforward Transformer for quasi-periodic pulse estimation in the spectral domain.

**Strengths:**

- Clear motivation and novelty:
The paper addresses a long-standing bottlenec: external illumination artifacts in a principled and physically grounded way using relative total variation.

- Strong quantitative and qualitative results:
The method outperforms competitive baselines across multiple datasets, with detailed ablations (e.g., β weight, STMap size, Gaussian σ) and computational cost analysis.

- Interpretability:
The disentanglement between illumination and physiological components provides visually intuitive frequency-space validation (t-SNE and PSD visualizations).

- Scalability and efficiency:
The model achieves high performance while maintaining competitive computational efficiency (≈6M parameters, 23 ms inference on RTX 4090).

**Weaknesses:**

- Limited generalization study:
The evaluation focuses primarily on facial datasets (COHFACE, BUAA-MIHR, MR-NIRP) without validation on diverse demographics, skin tones, or camera modalities.
For comparison, Wang et al., CVPR 2022 (Synthetic Generation of Face Videos with Plethysmograph Physiology) demonstrated strong generalization across diverse demographics and lighting; referencing this work would strengthen the empirical foundation.

- Dependency on precise face ROI segmentation:
The method assumes reliable ROI extraction; robustness under occlusion or misalignment is not evaluated.

- Complexity of self-supervised contrastive setup:
While effective, the sampling strategy (8×-12× rescaling for negatives) may require delicate tuning and is not well-justified theoretically.

- RTV parameter sensitivity:
Although ablation is provided, the RTV regularization parameter λ = 0.01 and σ = 5 are fixed; adaptivity across scenes could improve robustness.

- Limited temporal diversity testing:
The method is evaluated on 320-frame clips (≈12–15 s). It’s unclear how well it generalizes to real-time continuous streams.

**Questions:**

- Could you quantify how the RTV-based decomposition contributes to performance improvements compared to simpler spatial smoothing or attention-based decoupling?

- How sensitive is the performance to ROI extraction errors or background motion (e.g., moving cars, flashing lights)?

- Would the RTV-based filtering distort or attenuate high-frequency physiological details (e.g., HR variability)?

---

### Note · Authors · 2025-11-14

I have read and agree with the venue's withdrawal policy on behalf of myself and my co-authors.